# α-Amino Acids as Synthons in the Ugi-5-Centers-4-Components Reaction: Chemistry and Applications

**Sveva Pelliccia [1],\*, Ilenia Antonella Alfano [1], Ubaldina Galli [2], Ettore Novellino [1], Mariateresa Giustiniano [1],\*** and **Gian Cesare Tron [2]**

[1] Dipartimento di Farmacia, Università di Napoli "Federico II", Via D. Montesano 49, 80131 Napoli, Italy; alfano.ilenia@hotmail.it (I.A.A.); ettore.novellino@unina.it (E.N.)

[2] Dipartimento di Scienza del Farmaco, Università del Piemonte Orientale, Largo Donegani 2, 28100 Novara, Italy; ubaldina.galli@uniupo.it (U.G.); giancesare.tron@uniupo.it (G.C.T.)

\* Correspondence: sveva.pelliccia@unina.it (S.P.); mariateresa.giustiniano@unina.it (M.G.)

**Abstract:** Since the first reports, the Ugi four-component reaction (U-4CR) has been recognized as a keystone transformation enabling the synthesis of peptide mimetics in a single step and with high atom economy. In recent decades, the U-4CR has been a source of inspiration for many chemists fascinated by the possibility of identifying new efficient organic reactions by simply changing one of the components or by coupling in tandem the multicomponent process with a huge variety of organic transformations. Herein we review the synthetic potentialities, the boundaries, and the applications of the U-4CR involving α-amino acids, where the presence of two functional groups—the amino and the carboxylic acids—allowed a 5-center 4-component Ugi-like reaction, a powerful tool to gain access to drug-like multi-functionalized scaffolds.

**Keywords:** Multicomponent reactions; α-amino acids; Ugi four-component reaction; isocyanides

## 1. Introduction

Multicomponent reactions (MCRs) represent an efficient one-pot synthetic strategy to generate, from three or more reagents, a new product containing almost all portions of the starting materials. For their convergent nature, atom economy and efficiency, MCRs are considered valuable methodologies for both medicinal and organic chemists. In particular, Isocyanide-based Multicomponent Reactions (IMCRs) [1–4], have been proven to be an ideal tool to provide in one single step medium-complexity molecular skeletons, usually accessible only via a multistep approach. Most of MCR chemistry is performed with isocyanides and is related to the Ugi reaction [5,6], a four-component transformation (4CR) described in 1959 by Professor Ivar Ugi. In the reaction, an acid component, like a carboxylic acid, reacts with an oxo-component (a ketone or an aldehyde), a primary amine, and an isocyanide. In detail, as shown in Scheme 1, the first step is the condensation between the oxo-component **1** and the amine **2** to generate the Schiff base **6**. Then, the acid component protonates the nitrogen atom of the Schiff base thus increasing its electrophilicity. Hence, the nucleophilic addition of the isocyanide **3** to the Schiff base produces the nitrilium ion **9**, which rapidly reacts with the nucleophilic carboxylic acid anion **8**. The α-adduct **10** so formed is finally converted into the Ugi product **5** through an intramolecular acylation, which resembles the Mumm-type rearrangement.

**Scheme 1.** The Ugi 4-component reaction.

The replacement of one of the components of the Ugi reaction, or the incorporation of two or more functional groups into a single moiety, as well as the combination of the Ugi reaction with other chemical transformations have provided straightforward synthetic approaches to a large number of different scaffolds with a rich structural diversity. In particular, the presence of a primary amine and a carboxylic acid functional group both in natural and unnatural $\alpha$-amino acids make them very useful synthons suitable for a three-component Ugi-like reaction (3CUR) [7].

In this review, we summarized all the Ugi reactions involving $\alpha$-amino acids reported to date (March 2019). For the sake of simplicity, this review has been divided into four sections. In the second section, linear products will be gathered, while the third section will contain cyclic products obtained via the 3CUR. In the fourth section, we will focus our attention on products designed and synthesized for medicinal chemistry applications, while in the fifth section synthetic and biosynthetic reactions of $\alpha$-amino acids in Ugi-like transformations for the obtainment of natural products will be taken into consideration. For better clarity, in some cases, specific examples have been reported, while in others, d the general reaction is reported. Nevertheless, for any given transformation, the number of reported examples and the range of yields are always reported, when available.

## 2. Linear Compounds

### 2.1. Ugi 5C-4CR Using α-Amino Acids, Aldehydes, Chloroaldehydes and Ketones

In 1996, Ugi et al. described the first Ugi 5C-4CR using an $\alpha$-amino acid as a starting bi-functional material to yield 1,1'-iminodicarboxylic acid derivatives (97–99% yield, Scheme 2) [8–14]. In the proposed reaction mechanism (Scheme 2) the amine functional group of the $\alpha$-amino acid condenses with the aldehyde **1** to give the imine **12**. The subsequent $\alpha$-addition of the isocyanide followed by the intramolecular interception of the nitrilium ion, forms an *O*-acylamide **13**. The nucleophilic attack of the fourth component (the alcohol, i.e., the fifth reacting center) **14** at the carboxylic carbon atom and the subsequent rearrangement provides the 1,1'-iminodicarboxylic acid ester derivative **15**. Notably, the side chain of tri-functional $\alpha$-amino acids (*L*-Serine, *L*-Threonine, *L*-Tyrosine, *L*-Asparagine, *L*-Glutamine, *L*-Methionine) did not participate in the Ugi reaction.

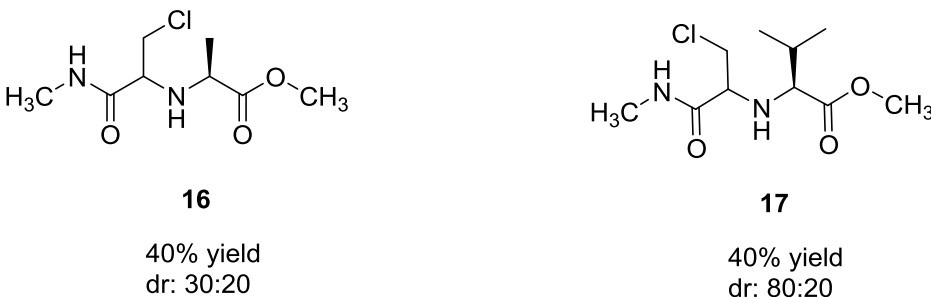

**Scheme 2.** The Ugi 5-center-4-components reaction.

Ugi et al. also investigated the use of a halogen-substituted aldehydes (Figure 1), in particular chloroacetaldehyde with methylisocyanide in methanol. Interestingly, only traces (up to 10%) of the substitution on halogen with the secondary amine of U-5C-4CR product were observed. Yields and diasteromeric excesses were less than those observed using unsubstituted aldehydes.

**Figure 1.** Use of halogen-substituted aldehydes with α-amino acids.

The U-5C-4CR was conducted also by using ketones instead of aldehydes [15]. Due to the low reaction rate of ketones, the reaction time increased from approximately one day to several weeks (4 examples, 64–94% yield). The same authors reported the formation of cyclic analogues, which are reported in Section 3.

### 2.2. Tandem Ugi-asserini Reactions Involving α-Amino Acids

The use of lysine triggered some interesting studies regarding the possibility of combining more than one MCR in tandem. Indeed, this α-amino acid can react with two equivalents of aldehyde, an isocyanide, and a carboxylic acid to afford the linear product **23** in 22.8% yield (Scheme 3) along with the Passerini product **24** in 18.5% yield [16]. Excess of isocyanide, and carboxylic acid led to the Ugi Nine Center Seven Component Reaction (U-9C-7CR) product **25** in 8% yield.

L-Lys  +  2 iPrCHO  +  2 tBuNC  +  MeCOOH

**18**      **19**      **20**      **21**

MeOH
**22**

U-5C-4CR     **23**

P-3CR     **24**

U-9C-7CR
**25**

**Scheme 3.** Use of *L*-lysine in the combination of MCRs.

Similarly, a tandem Ugi-Passerini reaction was observed when *L*- glutamic and *L*-aspartic acid were used (Scheme 4) [9]. In particular, the Ugi reaction using *L*-aspartic acid (**26**) with two equivalents of both aldehyde and isocyanide provided a linear compound **27** in contrast to from *L*-glutamic acid that gave cyclic diketopiperazine.

**26**    +    2    **19**    +    2    **20**    $\xrightarrow{CH_3OH}$    **27**

**27**

85% yield

dr: 50/27/13/10

**Scheme 4.** Use of *L*-aspartic acid in the combination of MCRs.

### 2.3. α-Amino Acids as Chiral Auxiliaries in the U-5C-4CR Reaction

Transtech Pharma patented the use of α-amino acids as chiral auxiliaries for the synthesis of enantiomerically pure *N*-alkyl-*N*-acyl-α-amino amides **15**, which after the cleavage of both the chiral auxiliary amine and the hydrolysis of amide, and the subsequent protection of the amino group, provided *N*-protected α-amino acids **30** (Scheme 5) [17].

**Scheme 5.** Use of α-amino acids as chiral auxiliaries in U-5C-4CR.

### 2.4. Diastereoselectivity in the U-5C-4CR Reaction

Sung, K. et al. investigated the steric effect of both aldehydes and α-amino acids on the diastereoselectivity of the Ugi reaction [18]. When bulky aldehydes (9-anthraldehydes, 2-ethylbutyraldehyde and isobutyraldehyde) were used in combination with *L/D* α-amino acids, the diasteroisomeric excess was 99%, while with less bulky aldehydes (such as benzaldehyde or *n*-butyraldehyde), the final products were obtained with a lower diasteroisomeric excess (43–79% de). Similarly, bulky enantiomerically pure *L/D* α-amino acids gave higher de (99%). The proposed mechanism showed that bulky substituents preferred to stay at equatorial positions of the six-membered Ugi intermediate to prevent serious *1,3-diaxal* and *'butan-gauche'* nonbonding interactions (Scheme 6).

Further information about the possibility of a substantial diastereoselectivity in the U-5C-4CR came from a report by Chen X. et al. The Ugi reaction of α-amino acids such as *L*-valine and *L*-serine, aromatic aldehydes, and isocyanides was employed for the synthesis of 1,1′-iminodicarboxylic acid derivatives, which were then used as key intermediates to synthesize tetrahydroisoquinoline compounds (Scheme 7) [19]. *L*-valine gave tetrahydroisoquinolin-4-ol **42** in nine reaction steps with high stereoselectivity. The configuration of the cyclic compound **42** was hence determined by NOESY NMR. Substituent at C-3 and C-4 resulted in a *cis* configuration as showed by correlation between H-3 (δ = 2.73 ppm) and H-4 (δ = 4.57 ppm). Consequently, the absolute configuration of C-1 and C-4 in **42** was determined to be *R* and *S*, respectively. Subsequently, Ugi product **41** from (*S*)-valine had the *S,R* configuration. This result could support a general rule regarding the stereochemistry of the similar Ugi reactions harnessing primary α-amino acids.

**Scheme 6.** Use of bulky aldehydes in combination with *L/D* α-amino acids.

**Scheme 7.** Synthesis of strained tetrahydroisoquinolines via a U-5C-4CR.

## 2.5. Use of TiCl$_4$ as Lewis Acid in the U-5C-4CR Reaction

Ciufolini and coworkers screened two Brønsted acids and thirteen Lewis acids to extend the scope of Ugi reaction involving α-amino acids and aromatic aldehydes (Scheme 8) [20]. Brønsted acids proved to be ineffective and harmful promoters. For example, trifluoracetic acid showed no improvement of the reaction and methanesulfonic acid gave no desired compound, perhaps due to polymerization or degradation of the isocyanide. In contrast, Lewis acids had beneficial effects on both yields and rates. TiCl$_4$ showed the best results (75–90% yield), which were not attributable to the HCl in situ release. The sense of diastereoinduction is (*S,S*), as confirmed by X-ray crystallography.

**Scheme 8.** Titanium (IV) chloride catalyzed U-5C-4CR with aromatic aldehydes.

### 2.6. Ketones and Secondary α-Amino Acids in the U-5C-4CR Reaction

To expand the molecular diversity of Ugi 5C-4CR, various symmetrical and asymmetrical ketones were combined with secondary α-amino acids [21]. Also in this case, catalytic amounts of $TiCl_4$ as Lewis acid proved to increase the reaction yield. The combination of bulky ketones with unbulky isocyanides gave the best yields, while unbulky ketones gave no significative difference in yields if combined with bulky or unbulky isocyanides (0–67% yield, Scheme 9). The diastereoselectivity of the U-5C-4CR may be sensitive to the nature of coupling reagents and reaction conditions. Following 2D-NMR studies on strained cyclic analogues, the authors noticed that the degree and the sense of diastereoinduction observed could not be easily rationalized.

In general, the stereochemical outcomes of this Ugi-5C-4CR variant depended on the structure of both the isocyanide and the ketone employed. These results indicated that it was not possible to draw general conclusions and that the mechanism and the diastereoselectivity of the Ugi-5C-4CR employing ketones is still a subject of debate.

**Scheme 9.** Use of ketones and secondary α-amino acids in U-5C-4CR.

### 2.7. Direct Conversion of Ugi Ester Scaffold to Amide

Dömling and co-workers investigated the one-pot conversion of the methyl ester function of the U-5C-4CR scaffold [22]. They described a one-pot amidation (24 examples, 15–82% yield, Scheme 10) from α-amino acid methyl esters under solventless conditions and at ambient temperature or using THF as solvent and several different amines, including aliphatic, heterocyclic, aromatic, and functionalized ones.

**Scheme 10.** One-pot conversion of the methyl ester of U-5C-4CR products.

## 2.8. Selenium α-Amino Acids in the U-5C-4CR Reaction

The same group reported a variation of the Ugi reaction using bifunctional selenium α-amino acids as starting materials to generate derivatives including methyl selenocysteine **48** and selenomethionine **49** (11 examples, 48–95% yield, Scheme 11) [23].

**Scheme 11.** Use of selenium α-amino acids in U-5C-4CR.

On the other hand, the use of diselenocysteine in the Ugi 5C-4CR (Scheme 12) gave low yield of the desired compound **55**, even when stirring the reaction for 24 h or refluxing for 12 h, most probably because of the poor solubility of the diselenide **54**.

**Scheme 12.** Use of diselenocysteine in Ugi 5C-4CR.

### 2.9. Isocyanocarbamates in the U-5C-4CR

Sello et al. reported the U-5C-4CR with isocyanocarbamates, α-amino acids and aldehydes [24]. It is worth noting that, when histidine was used as starting α-amino acid, the product was linear rather than cyclic as reported by Ugi (Figure 2). On the other hand, with glutamine, although a linear product was expected, the observation of a piperazinedione structure was indicative of a ring-closure due to intramolecular attack of the glutamine primary amide into the Ugi imino anhydride intermediate.

**Expected product**                    **Actual product**

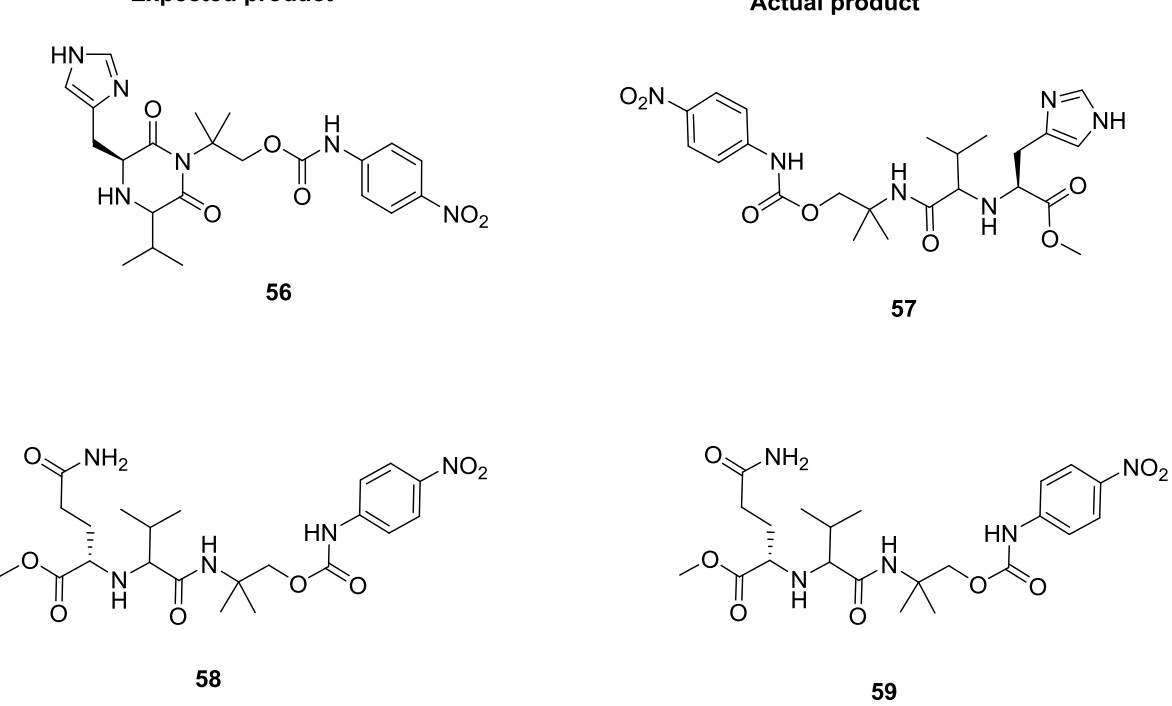

**Figure 2.** U-5C-4CR with isocyanocarbamates and histidine and glutamine.

### 2.10. Silica Nanoparticles as Green Catalyst for U-5C-4CR

Recently, Esrafili and coworkers described an efficient and green one-pot synthesis of sulfonylamide derivatives (42–68% yield, Scheme 13) from *L*-α-amino acids, aromatic aldehydes and *p*-toluenesulfonylmethyl isocyanide in water/methanol using silica nanoparticles (SNP) as the catalyst [25]. In the absence of SNP, the reactions did not work efficiently, and products **61** were obtained in low yields.

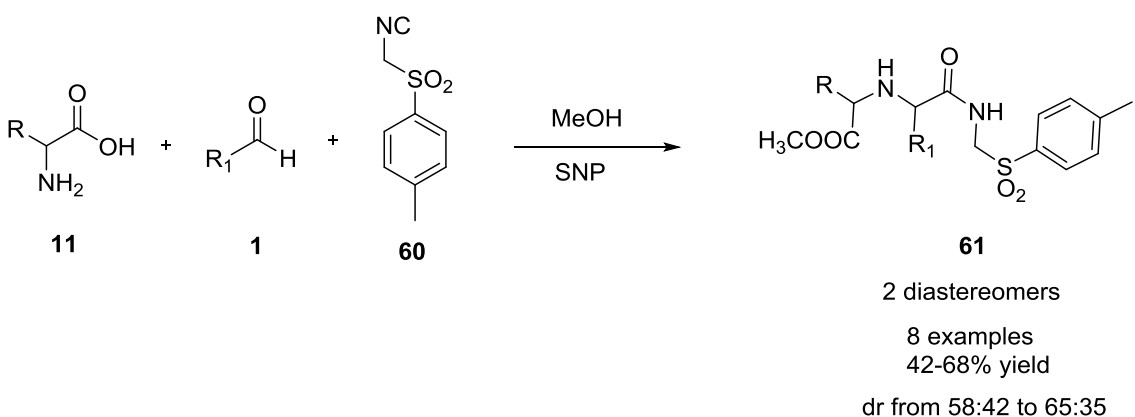

**Scheme 13.** SNP catalyzed synthesis of sulfonylamide derivatives.

### 2.11. Use of DMAP-Based Aldehydes in U-5C-4CR

Dimethylaminopyridine (DMAP)-based aldehydes were used with α-amino acids and *tert*-butyl isocyanide in the Ugi reaction to produce diverse chiral DMAP derivatives [26]. 4-(dimethylamino)-2-pyridine-carboxaldehyde **62** (Scheme 14) afforded products with low diasteroselectivity compared to 4-(dimethylamino)-3-pyridine carboxaldehyde **65** (Scheme 15). Diastereomeric ratios and yields were obtained changing substrates concentration and α-amino acids used in the Ugi reaction.

**Scheme 14.** U-5C-4CR with 4-(dimethylamino)-2-pyridine-carboxaldehyde.

**Scheme 15.** U-5C-4CR with 4-(dimethylamino)-3-pyridine carboxaldehyde.

The chiral diastereomerically pure DMAP derivatives were used for the kinetic resolution of secondary alcohols in presence of acetic anhydride and triethylamine in toluene [27].

## 3. Cyclic Compounds

### 3.1. Ugi-5C-4CRs and One-Pot Post-Condensation Modifications

One of the first reports regarding the obtaining of a diketopiperazine scaffold was by Ugi et al. in 1998. The formation of 2,6-piperazinediones was accomplished under weak basic conditions and in a one-pot reaction by using only ketones (43–70%, Scheme 16) [15]. Alternatively, the U-5C-4CR products **15** could be cyclized to 2,6-piperazindiones **68** by refluxing them in THF under basic conditions using potassium *tert*-butoxide (68–72% yield). However, stronger basic conditions and higher temperatures resulted in the racemization of the chiral centers.

One exception had been reported, again by Ugi, a couple of years previously, when histidine **69** was used as starting α-amino acid with an aldehyde and an isocyanoester. In this case, the formation of intramolecular H-bonds between the -*NH* of imidazole ring and the carbonyl group of isocyanoester **71** allowed for the isolation of the diketopiperazine derivative **72** when the reaction was run at room temperature and under neutral conditions (Scheme 17).

**Scheme 16.** Intramolecular attack leading to 2,6-piperazinediones.

**Scheme 17.** U-5C-4CR with histidine yielding diketopiperazine (**72**).

The sense of diastereoinduction, for reactions with unsymmetrical ketones, was studied by Turło et al. by converting the resulting Ugi adducts into the corresponding rigid 2,6-diketopiperazine derivatives (Scheme 18). In particular, in this study, the authors expanded the scope of the Ugi-5C-4CR to secondary $\alpha$-amino acids, such as proline, and to ketones. The linear adducts then underwent a *N*-detert-butylation/cyclocondensation sequence leading to *N*-unsubstituted cyclic derivatives **75, 76, 80–82** by reacting them in $BF_3 \cdots 2\,CH_3COOH$. Direct cyclization with NaOH of **83** and **86** gave directly cyclic compounds **85** and **87**. NOESY spectra of cyclic compounds showed that stereochemistry was not the same for all the synthesized compounds, but depended on the structure of both the isocyanide and the ketone inputs [21].

The use of siloxycyclopropanes in the Ugi-5C-4CR was described by Hans-Ulrich Reissig and coworkers to form highly substituted pyrrolidinone derivatives (Scheme 19) [28]. Siloxycyclopropanes **88** were used as direct precursor of $\beta$-formyl esters with *tert*-butyl isocyanide and $\alpha$-amino acids yielding the linear coupling products **89** (46–82% yield). These compounds were cyclized to pyrrolidinone derivatives **90** (16–98% yield) by heating them in toluene at reflux temperature.

When a chiral syloxycyclopropane was employed, the reaction resulted in 4 diastereoisomers; while non-chiral syloxycyclopropanes produced only 2 diastereoisomers (Scheme 20).

Precursors of $\gamma$-ketoesters such as **93** seemed to be inefficient in Ugi reactions, providing only 16% of the desired adduct **94** (Scheme 21).

Similarly, 1-trifluoromethyl-2-(trimethylsilyloxy)cyclopropanecarboxylate **95**, isonitriles and $\alpha$-amino acids (such as glycine **97** and phenylalanine **91**) provided $CF_3$-substituted $\gamma$-lactam (Scheme 22) [29]. In particular, the reaction with glycine at reflux temperature gave the cyclic compound **99** directly, without isolation of the linear compound **98**. On the contrary, when the U-5C-4CR was performed with phenylalanine, the reaction mixture was heating for 5 days to cyclize to $\gamma$-lactam **101**.

**Scheme 18.** Diketopiperazine derivatives useful to study diastero-induction of the U-5C-4CR.

**Scheme 19.** Reaction of siloxycyclopropanes with α-amino acids and isocyanides.

**Scheme 20.** Reaction of chiral siloxycyclopropanes with α-amino acids and isocyanides.

**Scheme 21.** Reaction of precursors of γ-ketoesters with α-amino acids and isocyanides.

**Scheme 22.** U-5C-4CR with 1-trifluoromethyl-2-(trimethylsilyloxy)-cyclopropane-carboxylate, isonitriles and α-amino acids.

The U-5C-4CR can be extremely powerful in getting access to molecular diversity and complexity when the linear adduct is further manipulated in a post-MCR transformation. A few examples have been reported herein. For example, highly functionalized constrained nitrogen-heterocycles were reported by Gracias et al. as examples of post-Ugi reaction manipulation [30]. Allyl glycine **103**, *o*-bromo-benzaldehyde **102** and benzyl isocyanide **52** in methanol yielded the Ugi cyclic adduct **104**, which was quickly converted into the aminoester **103**. Finally, a microwave-assisted Heck cyclization generated *N*-containing heterocycle (**106**) (90% yield, Scheme 23).

**Scheme 23.** U-5C-4CR and Heck post-MCR modifications.

Chiral highly functionalized dihydroisoquinolines and isoindoles were synthesized by Dyker and coworkers using U-5C-4CR followed by a gold-catalyzed hydroamination [31,32]. Dyker et al. used *L*-valine as the chiral amine component, and benzaldehydes with an alkyne moiety to generate highly functionalized secondary amines converted in dihydroisoquinolines **111** and **112** by a 6-*endo-dig* cyclization (23% and 35% yield) or in isoindoles **110** by a 5-*exo-dig* cyclization and subsequent aromatization such as in **113** and **114** (38% and 49%, Scheme 24). A further tandem Diels Alder reaction was carried on isoindole derivatives using acetylene dicarboxylic acid dimethyl ester **115**.

**Scheme 24.** U-5C-4CR and post-MCR modifications leading to dihydroisoquinolines and isoindoles.

The Pictet–Spengler reaction was recently applied in tandem as Ugi post-condensation transformation to yield complex polycyclic scaffold (Scheme 25). In these cases, the U-5C-4-CR adducts were reacted in one pot, without any purification, for the subsequent Pictet–Spengler cyclization [33–35]. This procedure was successfully carried out on differently functionalized Ugi scaffolds in order to obtain a wide range of multi-functionalized heterocycles. Isoindolone scaffold **119** was obtained by introducing methyl 2-formylbenzoate as the oxo-component and by monitoring the reaction via supercritical fluid chromatography-mass spectrometry (SFC-MS). Primary amines provided diastereomeric ratios of 70:30, and only the major diastereomer was precipitated during the reaction. In contrast, secondary amines gave four stereoisomers in equal ratios, suggesting racemization of the amino acid (Scheme 25).

**Scheme 25.** Isoindolones via U-5C-4CR and tandem Pictet–Spengler cyclization.

When β-ketoester, such as 2-oxocyclohexane carboxylic acid ethyl ester, was employed with 3 equivalents of cesium carbonate, pyrrolidinedione scaffold **121** was obtained either as a single diastereomer or mixture of two diastereomers easily isolated using preparative TLC plates (Scheme 26).

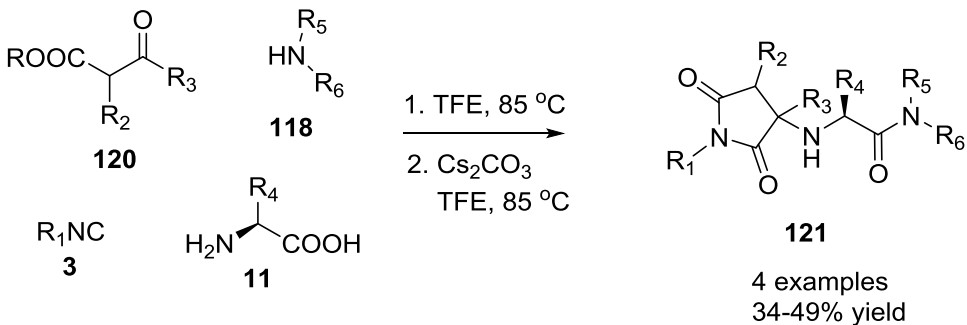

**Scheme 26.** Pyrrolidinediones via U-5C-4CR and tandem Pictet–Spengler cyclization.

Reaction involving *L*-triptophan as a starting α-amino acid provided the strained tricyclic 3,9-diazabicyclo[3.3.1]nonanes **124** in a one-pot two-step reaction, employing formic acid as the catalyst (Scheme 27).

Only U-5C-4C products bearing an electron-donating group on the *meta* position of the phenyl ring, such as 3-methoxy-phenylalanine and 3,4 dimethoxy-phenylalanine, were able to form isoquinolines derivatives **125** (Scheme 28).

**Scheme 27.** 3,9-Diazabicyclo[3.3.1]nonanes via U-5C-4CR and tandem Pictet–Spengler cyclization.

**Scheme 28.** Isoquinolines via U-5C-4CR and tandem Pictet–Spengler cyclization.

The use of electron-rich aromatic α-amino acids such as phenylalanine also gave bicyclic tetrahydroimidazo-[1,2-*a*] pyrazine-2,6(3*H*,5*H*)-diones **126** with the formation of only *trans* diastereomers (Scheme 29).

**Scheme 29.** Tetrahydroimidazo-[1,2-*a*] pyrazine-2,6(3*H*,5*H*)-diones via U-5C-4CR and tandem Pictet–Spengler cyclization.

L-leucine, cyclohexanone, benzyl isocyanide and morpholine gave an Ugi product whose formation was monitored through SFC-MS, subsequently reacted in the presence of one equivalent of potassium carbonate to provide β-lactam derivatives **130** (Scheme 30).

**Scheme 30.** β-lactam derivatives via U-5C-4CR and tandem Pictet–Spengler cyclization.

*3.2. Exploiting α-Amino Acids in the Combination of Tandem MCRs*

In theory, thanks to the presence of both the amino and the carboxylic acid functionality on the α-amino acids, and to the use of two equivalents of aldehyde and isocyanide, it could be possible to combine more than one MCR in the same flask. For example, glycine **97**, 2 equivalents of propionaldehyde **131**, 2 equivalents of methyl isocyanide **132**, and 1 equivalent of sodium azide **133** in methanol and in the presence of Dowex 50 allowed for a tandem Ugi-5C-4CR/ Ugi-azide 4CR, resulting in the formation in a single step of both tetrazole and 2,6-diketopiperazine rings **134** (Scheme 31) [16].

**Scheme 31.** Formation of tetrazole derivatives via tandem MCRs.

To enable a wide variability in the functionalization of the final product, the reaction could also be performed in two steps: equimolar amounts of α-amino acid **97**, aldehyde **131** and isocyanide **132** were one-pot mixed to give the Ugi-5C-4CR adduct **135**, which was not isolated, but reacted directly with a different aldehyde **19**, a different isocyanide **136** and sodium azide **133** to give a 1,1′-iminodicarboxylic acid derivative **137** (Scheme 32). It is worth noting that in this one-pot two-step approach, no cyclization was observed, and the linear adduct **137** was obtained.

**Scheme 32.** Formation of tetrazole derivatives via one-pot two-step tandem MCRs.

The possibility of performing more than one MCR in tandem, exploiting the amino and the carboxylic acid functionalities in two different reactions, has recently been investigated further. It was shown, indeed, that unprotected natural and unnatural α-amino acids, β/γ/ω-amino acids **138** with different aldehydes **1**, isocyanides **3** and sodium azide **133** furnished tetrazolo peptidomimetics **139** in good yields as diastereomeric mixtures (Scheme 33) [36]. Two sequential Ugi tetrazole/Ugi reactions were also performed to provide more complex structures **141**.

**Scheme 33.** Tetrazoles via tandem Ugi-tetrazole/U-4CR.

The Ugi tetrazole reaction also worked with dipeptides (Gly Gly) **143** and tripeptides (Gly Gly Gly) **144** (Scheme 34).

**Scheme 34.** Ugi-tetrazole reaction employing dipeptides and tripeptides.

The application of two or more one-pot tandem MCRs was also made possible by the use of equimolar amounts of sodium glycinate **149**, isocyanide **147**, and acetone **148** as the oxo-component and silver acetate as Lewis acid catalyst (Scheme 35) [37]. This reaction inserted a carboxylic acid function in the 2*H*-2-imidazoline **150**, which was then protonated and used in another Ugi 4-CR using *i*-PrCHO **19**, *n*-propylamine or benzylamine **2** and *tert*-butylisocyanide **20** to give compounds **151** and **152**.

Notably, this reaction was conducted in a one-pot sequence in which three different MCRs were combined to give the first example of an 8CR (Scheme 36). Extraordinarily, in this reaction, 5 new C-N bonds, and 4 new C-C bonds were formed in one pot.

**Scheme 35.** Two-step one-pot tandem U-5C-4CR/U-4CR.

**Scheme 36.** One-pot tandem U-5C-4CR/U-4CR.

On the other hand, as reported by Martens et al., when the α-amino acid was used in its hydrochloride form, the presence of the HCl served as the amine-protecting group, with only the carboxylic acid being able to be involved in an Ugi-like 4-CR [38,39]. In this reaction, the spiro derivatives of two 3-thiazolidines and one 3-oxazoline as imine component **157** were combined with glycine, β-alanine and γ-aminobutyric acid as hydrochloride salts **158**. Cyclohexylisocyanide, ethyl isocyanoacetate, *tert*-butyl isocyanoacetate, and *tert*-butyl isocyanide were employed to give a small library of oligopeptide analogues **159** (20–85%, Scheme 37).

**Scheme 37.** Spiro derivatives of two 3-thiazolidines and one 3-oxazoline as imine components.

### 3.3. Use of tri-Functional α-Amino Acids in Ugi-5C-4CRs

The use of α-amino acids with nucleophilic functional groups in their side chains such as amino-, sulfhydryl-, and hydroxyl-, enabled the synthesis of lactams, thiolactons, and lactones, respectively.

*L*-lysine **18**, for example, formed a ε-lactam **161** from the *O*-acylamide intermediate **160** via nucleophilic attack of the side chain amine to the ester (Scheme 38) [9].

**Scheme 38.** U-5C-4CR of *L*-lysine forming an ε-lactam.

A thiolactone scaffold **164** was generated during an Ugi 5C-4CR reaction using the α-amino acid homocysteine **162** with aldehydes and isocyanides in trifluorethanol as reported by Dömling et al. (Scheme 39) [40,41]. Trifluoroethanol was preferred as a solvent, as it did not react with the 6-membered α-adduct of the Ugi reaction, and because it favored the intramolecular reaction with the nucleophilic side chain of homocysteine.

**Scheme 39.** U-5C-4CR of homocysteine forming thiolactones.

Kim et al. illustrated an efficient synthesis for *N*-carbamoylmethyl-α-aminobutyrolactones **166** (42–97%, Scheme 40) starting from *L*-homoserine (**165**), aldehydes or ketones **1**, and isocyanides **3** in 2,2,2-trifluoroethanol (**163**) [42]. It is worth noting that the reaction with hindered aldehydes proceeded with high diastereoselectivity, probably due to steric factors.

When methanol was used as a solvent, the cyclic compound was obtained along with the ring-opened compound coming from methanol attack on the carboxylate carbon of the imino-anhydride intermediate **167** (Scheme 41).

**Scheme 40.** U-5C-4CR of homoserine affording α-aminobutyrolactones.

**Scheme 41.** Alternative formation of ring-opened compound **168**.

### 3.4. Combination of Other Bifunctional Starting Materials with α-Amino Acids

Kim et al. were able to demonstrate that commercially available glycolaldehyde dimer could be used efficiently in the Ugi condensation with α-amino acids to generate 3-substituted morpholin-2-one-5-carboxamide derivatives **170** (32–90% yield, Scheme 42) [43].

**Scheme 42.** Use of glycolaldehyde dimer in the U-5C-4CR.

When cyclic α-amino acids were used, other unique heterobicyclic compounds **172** (37–90% yield, Scheme 43) were produced in moderate to good yields.

**Scheme 43.** Use of glycolaldehyde dimer and cyclic α-amino acids in the U-5C-4CR.

The reaction mechanism forecasting the addition of the intramolecular hydroxyl group to the carboxylate carbon, resulting in the formation of 3-substituted morpholin-2-one-5-carboxamide derivatives **170** (Scheme 44).

**Scheme 44.** Reaction mechanism for the formation of morpholinones **170**.

The basic idea of the interception of the iminoanydride Ugi adduct by a hydroxyl group was also applied to the synthesis of macrolactones **178**. Cyclic chiral hemiacetals **176** and α-amino acids **11** combined with aliphatic, dipeptidic, glucosidic and lipidic isocyanides **3** in an Ugi 5C-3CR provided polysubstituted nine-membered ring lactones **178** (Scheme 45) [44]. The reaction led to poor diastereoselectivity in the formation of the new stereogenic center, but the complexity of the generated structures was remarkable.

**Scheme 45.** Formation of nine-membered lactones via U-5C-3CR.

Dömling et al. also described a variation of the previous reaction involving glycolaldehyde dimer, to form thiomorpholines, such as **180** and **181** (16–68%, Scheme 46), using α-amino acids **11**, mercaptoacetaldehyde **179**, and isocyanides **3** in trifluoroethanol as solvent [41,45]. Products were generally formed as a mixture of diastereomers that could be separated by silica gel chromatography. Cyclic α-amino acids, however, afforded mostly one diastereomer (Figure 3).

**Scheme 46.** Formation of thiomorpholines.

**182**
34%
dr: 76:24

**183**
47%
dr: 85:15

**184**
31%
dr: 64:36

**Figure 3.** Thiomorpholines formed with cyclic α-amino acids.

Novel glycopeptide structures were obtained by the extension of the Ugi reaction to an unprotected disaccharide with α-amino acids and isocyanides without any catalyst or reagent [46]. Boiling methanol, catalytic amounts of tertiary amines, and excess carbohydrate reduced reaction times and increased yields. *D*-ribose **185** was used with *D/L* α-amino acids **186** and ethyl isocyanoacetate **70** to provide a seven membered lactone such as **187** and **188** (Scheme 47). The α-amino acid isoelectric point influenced the reaction; neutral α-amino acids showed best results, while acidic or basic α-amino acids did not react. *L/D*-configured α-amino acids produced 1,2-*syn*- or 1,2-*anti*-configured seven-membered lactones, respectively, with a diastereoselectivity controlled by the steric demand of the α-amino acids employed. The configuration of the carbohydrates dictates the installation of the configuration at the carbon atom C-1 of the former carbohydrates. Disaccharides (maltose **189**, Scheme 48) or dipeptides (β-aspartame **192**, Scheme 49) were reacted under the same conditions.

**Scheme 47.** Use of *D*-ribose, *D/L* α-amino acids and ethyl isocyanoacetate.

**Scheme 48.** Reaction of disaccharides and *L*-proline.

**Scheme 49.** Reaction of dipeptides and *D*-ribose.

The reaction scope was explored with different pentoses and hexoses **194**, *L*-proline **190** or *D*-proline **195**, and ethyl isocyanoacetate **70** or toluenesulfonylmethyl isocyanide **60** (Scheme 50).

**Scheme 50.** Use of hexoses, *L/D*-proline, and ethyl isocyanoacetate.

Subsequently, this methodology was expanded to hydroxy ketones and dihydroxyketones with unprotected α-amino acids (Scheme 51) [47]. Reaction of hydroxyketones **198** and dihydroxyketones **200** with *L/D* α-amino acids **11** generated 2-oxo-morpholines **199** and **201** in good to high yields. Stereoselectivity is dictated by the nature of α-amino acids employed. *Syn*-configured oxomorpholines were detected preferentially.

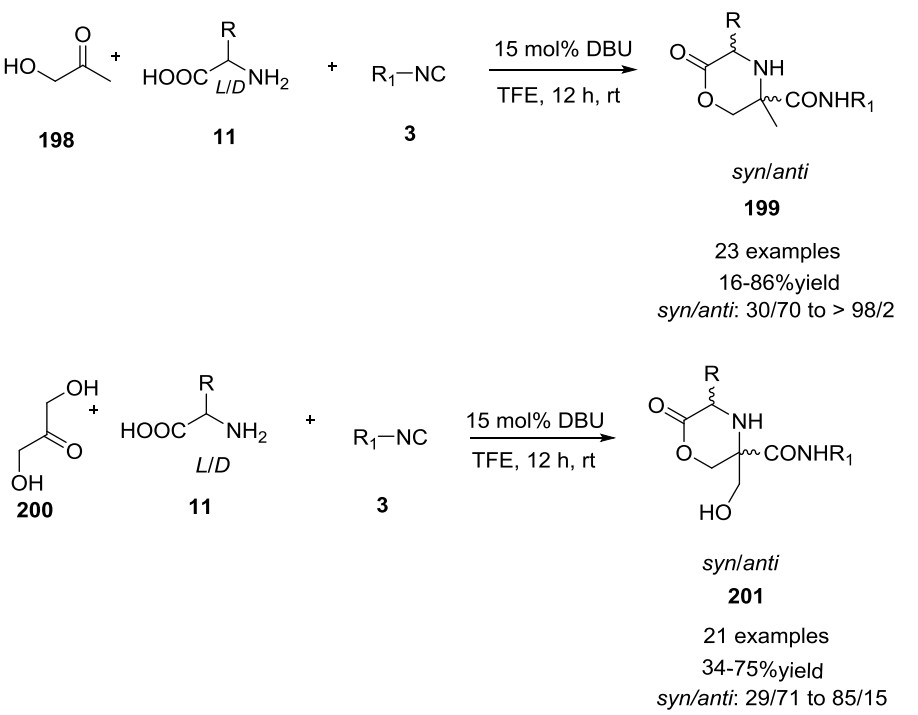

**Scheme 51.** Use of hydroxy- and dihydroxy-ketones.

Reaction of *L*-erythrulose (ketotetrose) **202** with *L*-valine (**38**) or *D*-valine **206** and *tert*-butyl isocyanide **20** in the presence of 1,5-diazabiciclo(5.4.0)undec-7-ene (DBU) gave different results depending on the *α*-amino acid stereochemistry (Scheme 52). In particular, the use of *L*-valine produced two different oxomorpholines **203** and **204** and a seven-membered lactone **205**, in contrast to *D*-valine, which gave a mixture of *syn-* and *anti*-configured 2-oxomorpholines **207**.

**Scheme 52.** Reaction of *L*-erythrulose (ketotetrose) with *L/D*-valine and *tert*-butyl isocyanide.

Ketohexoses showed different results in yield (Scheme 53). Only 60% of the yield of the *L*-valine series were observed when *D*-valine was used.

**Scheme 53.** Use of Ketohexoses with *L/D*-valine and *tert*-butyl isocyanide.

*L*-amino acids were used by Yudin A. et al. as starting material with isocyanides and aziridine aldehydes for piperazinone synthesis [48–50]. Piperazinone products **211** and **212** were obtained in

all cases as a single diasteroisomers without formation of linear product (Scheme 54). A detailed analysis through X-ray crystallography and NOESY NMR studies revealed that diastereoselectivity of piperazinones was dependent by the α-amino acid employed [51]. Primary α-amino acids gave a *trans* orientation to the new amino acid stereocenter **212**; in contrast, proline- and *N*-substituted α-amino acids formed *cis* product **211** (Figure 4). Achiral α-amino acids led to piperazinones with low diastereoselectivity.

**Scheme 54.** Use of aziridine aldehydes, isocyanides, and α-amino acids.

**Figure 4.** *Cis* and *trans* configurations of Yudin piperazinones.

The authors also investigated the role of the isocyanide in diastereoselectivity. Bulky isocyanides increased stereoselectivity, while electron-withdrawing groups on isocyanide decreased diastereoselectivity. The same trends were observed using bulky or unsubstituted aziridine aldehyde dimers. They also compared the reactivity and the stereoselectivity of diastereomeric *cis* and *trans* aziridine aldehyde dimers (Scheme 55). Using the *trans* methyl aziridine aldehyde dimer **213**, a moderate yield (up to 37%) was observed. In contrast, the *cis* methyl aziridine aldehyde dimer **215**, showed excellent diastereoselectivities, and moderate to good yields.

New synchronized synthesis of peptide-based macrocycles from three different components using a digital microfluidic platform was also presented [52]. The authors carried out the synthesis of a nine-membered macrocycle with an aziridine moiety in a fast, automated and well-controlled way. The system featured ten reagent reservoirs and eighty-eight actuation electrodes dedicated to dispensing, merging, and mixing droplets of reagents and products.

Solvatochromic fluorescent isocyanides **220** were also used in combination with aziridine aldehyde dimer **210** and α-amino acids **219** in the synthesis of environment-sensitive probes to obtain peptide macrocycles equipped with a fluorescent tag such as **221** (Scheme 56) [53]. Fluorophore macrocyclic peptides increased the mitochondria-localization compared to the linear one and could be used as irreversible probes of enzyme activity.

**Scheme 55.** Comparison between *cis* and *trans* aziridine aldehyde dimers.

**Scheme 56.** Use of fluorescent isocyanides to get macrocycles equipped with a fluorescent tag.

An environmentally friendly methodology for the synthesis of 6-alkyl/acyl phenanthridines such as **223** and **224** was recently reported by using unprotected α-amino acids as a source of stable alkyl/acyl radicals under metal free conditions with optimized reaction conditions (potassium carbonate and potassium persulfate as base and oxidant, respectively, Scheme 57) [54]. According to the proposed mechanism (Scheme 58), the homolytic cleavage of potassium persulfate generated sulfate radical anions **225** that gave single electron oxidative decarboxylation of α-amino acid anion **226**. This species was oxidized to iminium **227** and converted into aldehyde **1**. The sulfate radical anion abstracted the aldehyde hydrogen atom and this radical **228** could follow two different pathways. Direct decarboxylation could provide R radical **229** that reacted with isocyanide **230** to give an imidoyl radical **231**. This radical gave an intramolecular cyclization **232** and a subsequent oxidation to form the alkyl phenanthridine **223**.

The aldehyde radical could also react directly with isocyanide **230** to provide acylated phenanthridines **224**.

**Scheme 57.** Synthesis of phenathridines by using isocyanobiphenyls and α-amino acids.

**Scheme 58.** Proposed reaction mechanism for the formation of phenathridines.

Uyeda et al. reported the use of Pt/TiO$_2$ catalysts for light-induced dehydrogenation of methanol to formaldehyde **140** [55]. Upon excitation with UV light, TiO$_2$ is capable, when coupled with an efficient proton reduction catalyst such as Pt metal, of oxidizing alcohol substrates to aldehyde under mild dehydrogenation and less energetic visible/near-UV light illumination. H$_2$ was the only stoichiometric byproduct (Scheme 59).

In this report, a 100W Hg lamp was used for MeOH dehydrogenation to give formaldehyde **140** that was used in a Ugi reaction with unprotected *L*-proline **190**, 2,6-dimethylphenylisonitrile **234** in acetic acid to provide methyl ester product **235** in 54% yield (Scheme 60).

**Scheme 59.** Light-induced dehydrogenative imine transformations using Pt/TiO$_2$.

**Scheme 60.** Ugi reaction enabled by the photocatalytic dehydrogenation of MeOH.

## 4. Ugi Compounds in Medicinal Chemistry

The Ugi reaction, including the U-5C-4CR, has been extensively exploited for combinatorial diversity-oriented syntheses in the field of medicinal chemistry.

Peptide mimetic inhibitors of the P. falciparum M1 alanylaminopeptidase (APN), a key enzyme involved in malaria infection, were obtained by Ugi 5C-4C reaction. Gazarini et al. described an increase in both the number and the diversity of their 1,1'-iminodicarboxylic acid analogues produced by such multicomponent approach.

Previously [14] and newly synthesized [56] 1,1'-iminodicarboxylic acid analogues **236** were tested for PfA-M1 inhibition and for their in vitro antimalarial activity on the growth of P. falciparum erythrocytic stages (3D7 and FcB1 strains) (Scheme 61).

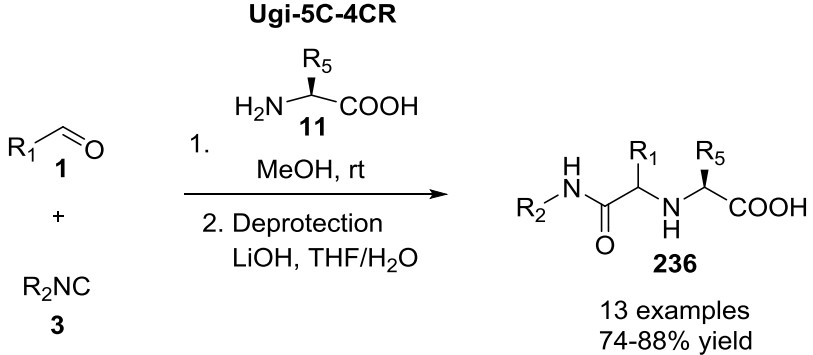

**Scheme 61.** Peptidomimetic inhibitors of the P. falciparum M1 alanyl-amino-peptidase.

The p53 protein has a key role in protecting cells with oncogenic mutations. It mediates growth arrest, senescence, and apoptosis in response to cellular damage. In normal cells, p53 is present in

very low levels because of its MDM-2 mediated degradation. So MDM2 represents the principal antagonist of p53 by limiting the p53 tumor suppressor function. It is structurally and biologically well understood that the key for p53–MDM2 interaction is a triad of p53 α-amino acids that inserts itself into the MDM2 cleft: Phe19, Trp23, and Leu26. For this reason, the design of molecules that inhibit the interaction of p53 and MDM2 could provide new therapeutic strategies for cancer.

Inhibitor KK271 (**239**) of the MDM2–p53 interaction based on the 6-chloroindole scaffold was synthesized through an U-5C-4CR using *L*-leucine **237**, ethyl 6-chloro-3-formyl-1*H*-indole-2-carboxylate **238** and benzyl isocyanide **52** (Scheme 62) [57].

6-Chloroindole-2-hydroxamic acid of KK271 fitted into the Trp23 pocket of MDM2, the isobutyl element filled the Phe19 pocket, and the benzyl moiety was bound within the Leu26 pocket. Analysis of the KK271-MDM2 crystal complex revealed the similarity with the native MDM2-p53 structure and the possibility for further modifications on the central, peptidic core in order to improve the drug likeness.

**Scheme 62.** Inhibitor KK271 of the MDM2–p53 interaction.

High-voltage-activated $Ca^{2+}$ channels are involved in contraction, secretion, neurotransmitter release, and gene expression and, as shown in many reports, T-type $Ca^{2+}$ channel blockers are crucial for treatment of epilepsy and neuropathic pain.

Morpholin-2-one-5-carboxamide derivatives, such as **240** and **241**, synthesized via an Ugi-5C-4CR described by Kim et al. were tested as a novel class of potent and selective T-type $Ca^{2+}$ channel blockers (Scheme 63) [58]. They were preliminarily screened against CaV3.2 T-type $Ca^{2+}$ channels expressed in Xenopus oocytes. Compounds exhibiting more than 45% inhibition were re-evaluated for the blocking effects on CaV3.1 channels expressed in HEK293 cells. Usually, 3,5-*cis* adducts **240** showed higher activity than their *trans* analogues **241** and selective effects on T-type channels compared with N-type channels.

**Scheme 63.** Morpholin-2-one-5-carboxamides as T-type $Ca^{2+}$ channel blockers.

Substantial need for new, more effective and safer anticonvulsant drugs with lower side effects and drug–drug interactions, and with the challenge of being *disease modifying*, prompted Turlo et al. to Test 2,6 diketopiperazines previously synthesized via U-5C-4CR in various animal models of epilepsy.

2,6-Diketopiperazines were synthesized, starting from non-polar α-amino acid **186** (*L*-valine, *L*-leucine, *L*-isoleucine, *L*-phenylalanine, *L*-phenylglycine), benzaldehyde **142**, *tert*-butyl isocyanide **20**, and methanol in the presence of a catalytic amount of Iron (III) chloride (Scheme 64) [59]. *Tert*-butyl cleavage by use of $BF_3 \cdot CH_3COOH$ and base-induced intramolecular cyclocondensation gave the final products **244** and **245**, which displayed a good anticonvulsant activity in various animal models of epilepsy. Structure–activity relationship studies highlighted all the requirements for the anticonvulsant activity: proper stereochemistry on the stereogenic centers (*S*,*S*), the presence of an imide moiety and a benzene ring attached to 2,6-DKP scaffold. They also analyzed less sterically constrained monocyclic piperazines by removing the second condensed ring derived from *L*-proline or *L*-homoproline, thereby better fitting into the putative receptor(s). Synthesized compounds showed weak to good anticonvulsant activities in maximal electroshock seizure tests.

A similar approach was used by Glaxo to synthesize linear compounds subsequently cyclized into diketopiperazines that had high affinity as antagonists of the oxytocin receptors on the uterus of rats and humans [60]. DKPs exhibited also high affinity at the human recombinant oxytocin receptor in CHO cells.

**Scheme 64.** 2,6-Diketopiperazines with anticonvulsant activity.

## 5. UGI-5C-4CR in the Synthesis of Natural Products

The development of new synthetic strategies of terpene isocyanides found in marine organisms has represented a crucial investigation over recent decades.

Boneratamides A–C were isolated from the marine sponge *Axinyssa aplysinoides* by Andersen and co-workers in 2004. A retrosynthetic approach showed a simple synthetic strategy through the use of three building blocks (axisonitrile-3 **246**), a carbonyl component, either acetone or acetaldehyde **148** or **247**, and glutamic acid **248** (Scheme 65) [61,62].

**Scheme 65.** U-5C-4CR leading to boneratamides A–C.

Hypothetical biosyntheses of the right-side portion of boneratamide A **252** and **253** (Scheme 66) and exigurin **283** (Scheme 67) were proposed by Yoshiyasu and coworkers according to an U-5C-4CR. Based on these proposals, a biomimetic approach was designed and applied to the synthesis of these molecular frameworks.

**Scheme 66.** Biomimetic approach for the synthesis of boneratamide A.

**Scheme 67.** Biomimetic approach for the synthesis of exigurin (right side).

To exclude mixtures of γ-lactams, boneratamide B–C right-side portion syntheses were conducted through a stepwise synthetic route (Scheme 68).

**Scheme 68.** Stepwise synthetic route to boneratamide B–C.

Synthesis of the despirocyclic boneratamide A analogue **263** was also explored by preparing the isocyanide **262** from (+)-menthol **260** via azide formation **261**, hydrogenation to amine, formamide formation and final dehydration (Scheme 69).

**Scheme 69.** Synthesis of the despirocyclic boneratamide A.

Halichonadin G is a natural marine product isolated in 2011 by Kobayashi and co-workers from the sponge *Halichondria* sp. Yoshiyasu and coworkers hypothesized the natural synthesis of Halichonadin G from a natural precursor, i.e., Halichonadin C, considering a putative "ugiase" that might promote a Ugi reaction (Scheme 70) [63]. In contrast to Boneratamide A–C, the synthesis of Halichonadin G using *N*-unprotected α-amino acids was unsuccessful, giving only undesired products. However, the use of *N*-benzylglycine allowed biomimetic U-5C-4CR, a menthyl analogue of Halichonadin G.

The syntheses of the right-hand portion of Halichonadin Q and the central part of Halichonadin M have also been reported [64].

The Kobayashi group isolated Halichonadins M–Q, from the light-brown marine sponge *Halichondria* sp. collected at Unten Port in Okinawa Island. Like other terpenes previously described, the natural synthesis could start from Halichonadin C isocyanide **264** by a putative "ugiase" (Scheme 71).

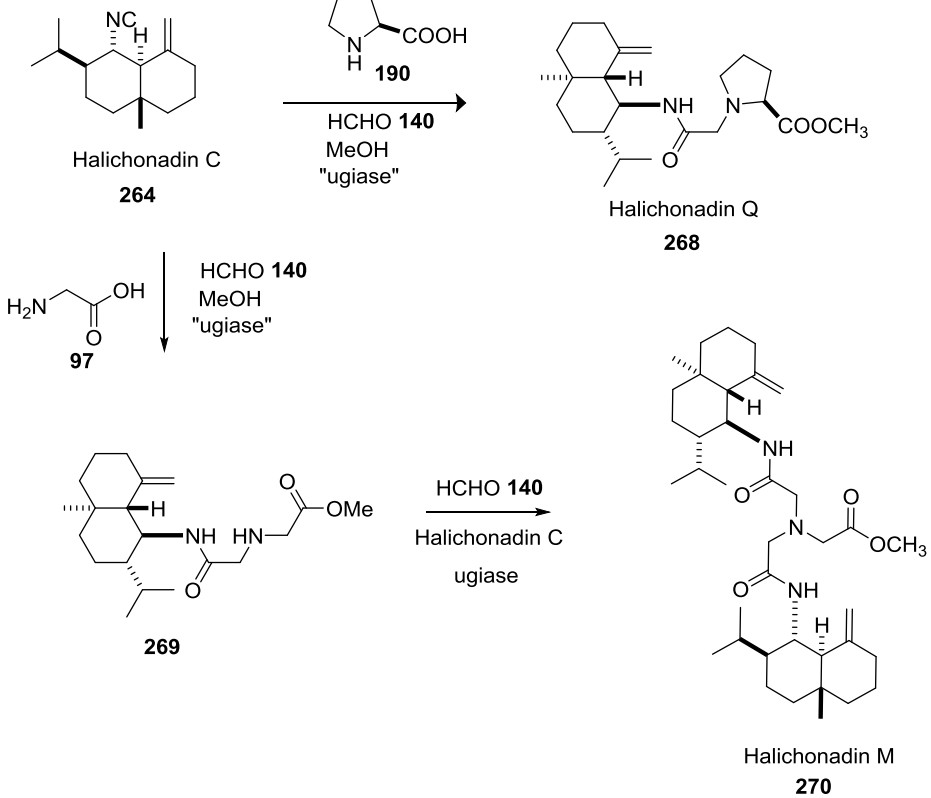

**Scheme 70.** Natural synthesis of Halichonadin G promoted by a putative "ugiase".

**Scheme 71.** Plausible biosynthetic pathway of Halichonadin M.

A biomimetic one-pot process could also be applied for the synthesis of these terpenes **271** and **272** in a simple and remarkable manner (Scheme 72).

Scheme 72. Biomimetic one-pot process for the synthesis of halichonadin analogues.

Analogously, the syntheses of despiro analogues of exigurin **274** (Scheme 73) and boneratamide B–C (Scheme 74) were explored using the same U-5C-4CR [65].

Scheme 73. Synthesis of despiro analogue of exigurin.

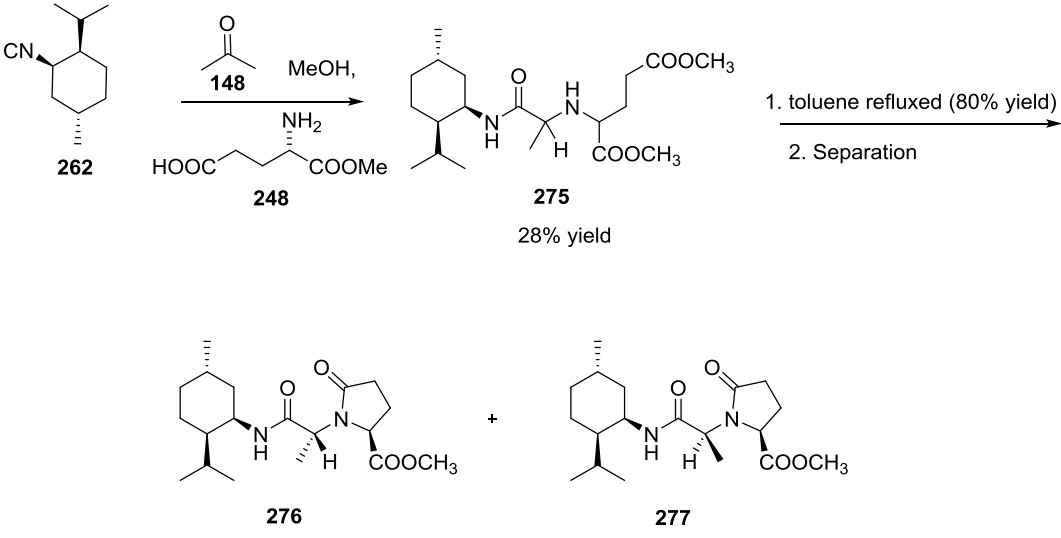

Scheme 74. Synthesis of boneratamide B–C analogues.

## 6. Conclusions

As highlighted by this review, the U-5C-4CR has been exploited for the synthesis of a wide range of organic compounds, including peptide mimetics, heterocycles, and natural compounds. Notably, its potentialities have also been exploited by Yudin et al. in the macrocyclization of peptides, which is often a difficult task for organic chemists facing long-standing and unsolved problems affecting yields and purity of cyclic peptides. Despite being a 70-year-old reaction, we believe that the potentialities of the U-5C-4CR have still to be fully discovered, and we hope that this collection of literature reports will be useful in renewing the never-ending interest in such efficient processes and triggering new relevant applications in all the fields of chemistry.

**Author Contributions:** Conceptualization, S.P., M.G., G.C.T.; Writing-original draft preparation, S.P., I.A.A., U.G.; Writing-review and editing, S.P.; M.G.; G.C.T.; E.N.

**Funding:** This research received no external funding.

**Acknowledgments:** Financial support from Università degli Studi "Federico II" Napoli, Italy and Università del Piemonte Orientale, Novara is acknowledged. S.P., I.A.A., and M.G. gratefully acknowledge MFAG 18793.

**Conflicts of Interest:** The authors declare no conflict of interest.

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
