# Peer review of "α-Amino Acids as Synthons in the Ugi-5-Centers-4-Components Reaction: Chemistry and Applications"

_symmetry, doi:10.3390/sym11060798_

Round 1

Reviewer 1 Report

The paper summarizes an intriguing substrate class (amino acids) in the context of several useful multi-component coupling reactions. The authors comprehensively describe the literature in this vibrant niche of the chemical literature. Numerous interesting examples are documented, as well as instances where side-products are observed. The experimental results (yield, stereoselectivity) are described satisfactorily.

The manuscript could be greatly improved with additional organization. The document is separated into 4 sections (liner, cyclic, medicinal compounds, natural products). However, the investigation of linear adducts is extensive and the results are not grouped by substrate class, the number of components, or product outcome. Additional subheadings or organization is needed for the reader to be able to quickly identify the pertinent portion(s) of the review. 

Furthermore, the discussion of examples in the medicinal chemistry (Sec. 4) and natural products is very short and superficially redundant with the preceding section(s). The fact that the discussion of the MCR's in medicinal chemistry is only two pages and that of natural products is 7 pages, isn't due to a lack of interesting chemistry to describe. Instead, it appears the authors have little insight into these articles providing only 1-2 sentences per example and 2-3 sentences per page. These sections (4 and 5) should have additional detail added.

Items that would improve clarity for the reader:

p2 Scheme 1 Change Mumm to Mumm Type Rearrangement, conversion of 9 to 5 should indicate the intermediate tetrahedral intermediate or omit arrows so as to not deceive the reader.

p3 scheme 2 The authors should use dr since the ratio reflects the relative rates or equilibrium constant rather than de. J Org Chem. 2006 Mar 17; 71(6): 2411–2416.

p7 scheme 7 Include labels for "H-3" and H-4" to improve clarity for the reader.

p12 scheme 16 image has been distorted or does not follow template

p13 scheme 18 This scheme shows more detail, in particular not related to the MCR, but rather the stereochemical analysis than the article does elsewhere. In addition, the description of this reference is lacking as compounds 71-85 are not eluded to in the text.

p14 scheme 19/20 There is a numbering issue #88 is used twice. #87 is used for two different compounds.

p18-19 References 33-35 are not appropriately summarized for the reader. The single paragraph text from these 4 articles are shown in 6 schemes. It is difficult for the reader to garner the important outcomes and limitations found in these references.

p31 Scheme 54 utilizes beta-amino acids which are not indicated to be the focus of the manuscript.

p33 1 scheme 57 MCR once again without alpha-amino acids which appears to be outside the scope of the review.

p35 scheme 61 utilizes a coloring scheme that is inconsistent with all other figures in the manuscript.

p38 Inconsistent margins are present in paragraph 1

p38 scheme 67 label is not formatted correctly

Untraditional Word choices that could be improved:

p1 l16 "whatever organic transformation"

p1 l24 "precious"

p1 l25 demonstrated -> proven

p1 l32 -> increasing the electrophilicity of 7.

p2 l45 "made" -> "make"

p2 l56 "amino function" -> amine functional group

p7 l145 "not possible to drawn general conclusions"

p7 l169 "It is worth of note that,.."

p7 l170 "albeit" -> although

p8 l183 DMAP is not defined

p28 DBU is not defined

p32 l490 "The authors analyzed also the..."

Author Response

Point by point response reviewer  1

1. The paper summarizes an intriguing substrate class (amino acids) in the context of several useful multi-component coupling reactions. The authors comprehensively describe the literature in this vibrant niche of the chemical literature. Numerous interesting examples are documented, as well as instances where side-products are observed. The experimental results (yield, stereoselectivity) are described satisfactorily.

The manuscript could be greatly improved with additional organization. The document is separated into 4 sections (liner, cyclic, medicinal compounds, natural products). However, the investigation of linear adducts is extensive and the results are not grouped by substrate class, the number of components, or product outcome. Additional subheadings or organization is needed for the reader to be able to quickly identify the pertinent portion(s) of the review. “

R: We grouped linear product section adding subheadings

2.” Furthermore, the discussion of examples in the medicinal chemistry (Sec. 4) and natural products is very short and superficially redundant with the preceding section(s). The fact that the discussion of the MCR's in medicinal chemistry is only two pages and that of natural products is 7 pages, isn't due to a lack of interesting chemistry to describe. Instead, it appears the authors have little insight into these articles providing only 1-2 sentences per example and 2-3 sentences per page. These sections (4 and 5) should have additional detail added.”

R: Additional details were provided in Section 4.

3. “p2 Scheme 1 Change Mumm to Mumm Type Rearrangement, conversion of 9 to 5 should indicate the intermediate tetrahedral intermediate or omit arrows so as to not deceive the reader.”

R: We changed Mumm to Mumm type rearrangement in both text and scheme.

4.” p3 scheme 2 The authors should use dr since the ratio reflects the relative rates or equilibrium constant rather than de. J Org Chem. 2006 Mar 17; 71(6): 2411–2416.”

R: As suggested by reviewer and described in J Org Chem. 2006 Mar 17; 71(6): 2411–2416 we used dr rather than de.

5. “p7 scheme 7 Include labels for "H-3" and H-4" to improve clarity for the reader.”

R: We highlighted H3 and H4 in Scheme 7

6.” p12 scheme 16 image has been distorted or does not follow template”

R: We modified Scheme 16

7. “p13 scheme 18 This scheme shows more detail, in particular not related to the MCR, but rather the stereochemical analysis than the article does elsewhere. In addition, the description of this reference is lacking as compounds 71-85 are not eluded to in the text.”

R: We described cyclic compounds with their numbers in the text

8. “p14 scheme 19/20 There is a numbering issue #88 is used twice. #87 is used for two different compounds.”

R: We modified numbers starting from the beginning , as also suggested by reviewer 2

9.” p18-19 References 33-35 are not appropriately summarized for the reader. The single paragraph text from these 4 articles are shown in 6 schemes. It is difficult for the reader to garner the important outcomes and limitations found in these references.”

R: We added a short paragraph with comments and remarks for all the schemes referred to the Pictet Spengler reaction.

10.” p31 Scheme 54 utilizes beta-amino acids which are not indicated to be the focus of the manuscript.”

R: We removed Scheme 54.

11.” p33 1 scheme 57 MCR once again without alpha-amino acids which appears to be outside the scope of the review”

R: We removed Scheme 57, 58, 59 and 61 and also peptide from Scheme 60.

12.” p35 scheme 61 utilizes a coloring scheme that is inconsistent with all other figures in the manuscript”

R:  Scheme 61 described macrocycles from RGD peptides. Following the previous suggestion we decided to remove this Scheme.

13.” p38 Inconsistent margins are present in paragraph 1” ; “p38 scheme 67 label is not formatted correctly”

R: In our PDF file margin and label are normal. Probably there was a problem in file generation during the submission procedure.

Untraditional Word choices that could be improved:

R: The following words have been replaced:

p1 l16 "whatever organic transformation" changed with “a huge variety of”

p1 l24 "precious" changed with valuable

p1 l25 demonstrated -> proven

p1 l32 -> increasing the electrophilicity of 7.

p2 l45 "made" -> "make"

p2 l56 "amino function" -> amine functional group

p7 l145 "not possible to drawn general conclusions" converted with to draw

p7 l169 "It is worth of note that,.."changed with It’s worth noting

p7 l170 "albeit" -> although

p8 l183 DMAP is not defined.  R: as suggested, We defined DMAP in the text

p28 DBU is not defined. R: DBU has been indicated with its IUPAC in name in text.

p32 l490 "The authors analyzed also the..." changed with The authors investigated also the

Reviewer 2 Report

Ugi four component reaction is quite interesting and important reaction in organic and medicinal chemistry field.

The reaction provides complexity and diversity in atom economic way.

this review quite well summarized this field.

linear compound,  cyclic compound, medicinal chemistry, natural product synthesis part was well organized.

some minor typo  was checked in attached file.

please review it and change if it's appropriate.

there are lots of missing character for example α, β, γ.

please modify it.

compound number 9 in second page is wrong and renumber the other compound.

de or d.r. need to be used in consistency. choose between them.

4 ℃, 4 h, 4 min, r.t., de, d.r. (these need to be used in consistency; this way or the other way.)

Please check the journal format.

in scheme 18 check CH2Ts or CH2OTs

line479 L-Amino acids

Author Response

Point by point reviewer 2

“some minor typo  was checked in attached file. please review it and change if it's appropriate.”

R: 1. “In one- single step” converted into “in a single step”

    2. “electrophilicity 7” converted in “electrophilicity”

    3. numbering changed from scheme 1 and also a, b, g etc.

    4. eliminated bold in the sentence

    5. structures in scheme 6 were corrected but sterochemistry of minor compound is shown as reported in the original paper

   6. the sentence in which there was efficacy was eliminated from the text

   7. “it was not  possible to drawn” corrected in “it was not possible to draw”

   8. we eliminated “very” in “very different amines” sentence

  9. 24h and 12h corrected with 24 h and 12 h

   10. we changed piperidindione in piperazinedione

  11. we adjusted scheme 16

  12. we corrected “syloxycyclopropane” with “siloxycyclopropane”

  13. we corrected “glicine” with “glycine”

  14. we changed “the authors” with “Dyker et al.”

  15. we changed “propionic aldehyde” with “ propionaldehyde”

  16. we cconverted again “in one single step” into “in a single step”

   17. “iPr” changed with “iPr”

   18. Scheme 34 caption modified adding “and tripeptides”

   19. we corrected  scheme 35 with tBu

   20. we changed “were one-pot formed” into “ formed in one pot”

   21. Scheme 42 and 43 corrected “-40oC- rt”  with ““-40oC to rt”

   22.  we changed “hydroxyl-“ into “hydroxyl group”

   22.Scheme 45. Sterochemistry of R2 is S as the starting aminoacid

   23. Scheme 46. Inserted de (73:27 to 96:4)

   24. Scheme 47: changed sterochemistry of N-C1

25. All the caption in Schemes are not in the centre; it is a problem of file conversion

26. Scheme 51: we inserted ratio syn/anti

27. Scheme 54 was eliminated as suggested by reviewer 1 because b-aminoacids are out of the topic

28. we rewrote “a-amino acid dependent diasteroselectivity of piperazinones” using “that diastereoselectivity of piperazinones was dependent by the a-aminoacid employed”

29. “ iBu” changed with  “iBu”

30. “aldeyde” corrected with “aldehyde”

31. scheme 57 and 58 were eliminated from review as suggested by reviewer 1

32. “moyety” corrected with “moiety”

33. In the actual Scheme 58 (ex 63) we added R2 and R3

34. In the actual Scheme 60 (ex 65) we placed MeOh on the top

35. we specified in vitro assay on the growth of P. falciparum erythrocytic stages (3D7 and FcB1 strains)

36.  “chloroindole”  changed with “Chloroindole”

37. Scheme 62 (ex 67) was not in a different size. It’s a conversion problem

38. Scheme 63 (ex 68) was not in the center like the others. . It’s a conversion problem

39. “2,6 diketopiperazines” converted into  “2,6-diketopiperazines”

40. “tert-butyl” changed in “tert-butyl “

41. eliminated 1 and 2 in bold in Scheme 64 (ex 69) and corrected “4,6 h” with “4-6 h”

42. Scheme 74 (ex 79) “Toluene” changed with “toluene”

“there are lots of missing character for example α, β, γ.” please modify it.

R: We are sorry, but these typos arose in the conversion of our word file into pdf format during the submission process.

“compound number 9 in second page is wrong and renumber the other compound.”

R: we renumbered from scheme 1

“de or d.r. need to be used in consistency. choose between them.”

R: We converted de into dr as suggested also by the first reviewer.

 “4 , 4 h, 4 min, r.t., de, d.r. (these need to be used in consistency; this way or the other way.)”

R: we choose one  way to express these values

“in scheme 18 check CH2Ts or CH2Ots”

R: we indicated CH2Ts as the authors did  In the article

“line479 L-Amino acids”

R: we corrected L- Aminoacids into L- Amino acids

Round 2

Reviewer 1 Report

The manuscript describes an intriguing substrate class (amino acids) in the context of multi-component coupling reactions. The authors comprehensively describe the literature in this vibrant niche of the chemical literature. Numerous interesting examples are documented, as well as instances where side-products are observed. The experimental results (yield, stereoselectivity) are described satisfactorily.

The revised manuscript greatly improved the organization of the review. Notably, Section 1 (linear substrates) was modified to include subheadings to help the reader to be able to quickly identify the pertinent portion(s) of the review. In addition, the discussion of the medicinal chemistry and naturals products (sections 3 and 4), was expanded to give the reader adequate insight into the impact and importance of the papers discussed. 

The authors responded to and corrected this reviewer's English grammar and typographical errors.

 Scheme 21 - incorrect symbol [?]